# REMIX: Resolution Enhancement through Mixture of Experts

**Sergio Morell-Ortega**[*1]                                SERMOOR1@TELECO.UPV.ES

[1] *Instituto de Aplicaciones de las Tecnologías de la Información y de las Comunicaciones Avanzadas (ITACA), Universitat Politècnica de València, Camino de Vera s/n, 46022, Valencia, Spain*

**Marina Ruiz-Perez**[1]                                MARINA.RUIZPEREZ11@GMAIL.COM

**Pierrick Coupé**[2]                                PIERRICK.COUPE@GMAIL.COM

[2] *CNRS, Univ. Bordeaux, Bordeaux INP, LABRI, UMR5800, in2brain, F-33400 Talence, France*

**José V.Manjón**[*1]                                JMANJON@FIS.UPV.ES

**Editors:** Under Review for MIDL 2024

## Abstract

We propose a novel method to enhance the resolution of magnetic resonance images (MRI) using deep learning. Our approach is based on realistic MRI data degradation using real affine registrations. We demonstrate the efficacy of a Mixture of Experts approach in handling diverse input resolutions commonly present in clinical settings. This is done by training seven networks, each one for a specific native volume resolution, allowing more effective handling of low-resolution images when compared to a unique model.

**Keywords:** Super-resolution, MRI, Deep Learning, Mixture Of Experts

## 1. Introduction

MR images are acquired at diverse resolutions worldwide. Efforts to bridge the gap between clinical and research scans have focused on super-resolution techniques, particularly modern deep Convolutional Neural Networks (CNNs) like SynthSR (Iglesias et al., 2021) and SMORE (Remedios et al., 2023). However, the challenge lies in acquiring low/high resolution (LR/HR) paired training data or simulating realistic degradation models and creating a model that can handle highly diverse resolution patterns. These challenges led to the development of our proposed method. We advocate for realistic data generation of LR data by leveraging affine registration between HR images from research environments (e.g., HCP (Van Essen et al., 2012)) and real clinic scans from the volBrain database.

## 2. Materials and methods

The proposed method for resolution enhancement in MRI images was developed using multiple datasets. The first dataset consisted of T1 images from the Human Connectome Project (HCP) database, with 75 randomly selected subjects. These images had dimensions of $260 \times 311 \times 260$ voxels and a resolution of $0.343 \text{ mm}^3$. Another dataset, containing 22,901 images, was obtained from the volBrain database[1] with a wide range of resolutions, including differences in slice thickness, intra-slice resolution, orientation, age, sex, and origin.

---

[*] Contributed equally

1. volBrain.net is an online neuroimaging analysis platform with a diverse database of over 300,000 images from around the world, shared by users who gave permission for research purposes.

A third dataset, sourced from various publicly available datasets (compiled in a lifespan study (Coupé et al., 2017)), was used to assess the method's performance.

Realistic data generation involves registering cases from the HCP and `volBrain` dataset to the MNI152 template using the ANTS registration tool (Avants et al., 2009). This step allows the creation of MNI space images with dimensions of 181x217x181 voxels and a voxel size of 1 mm$^3$. After registering the high-resolution HCP volumes to the MNI152 space, the inverse transform of a randomly chosen `volBrain` case is applied to simulate LR acquisition in the native space. Finally, the generated HCP image is returned back to the MNI152 space, containing realistic resolution/orientation-specific artefacts allowing for the simulation of a realistic LR case for a specific HR case (see Fig.1).

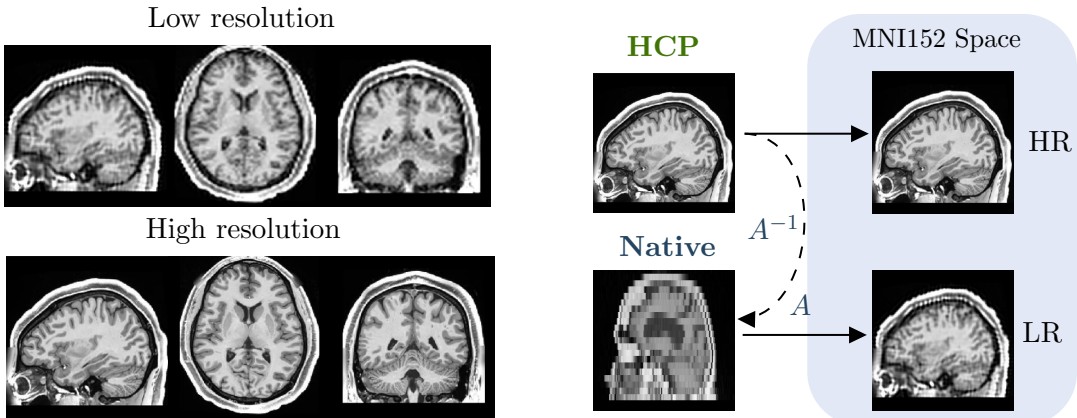

Figure 1: Realistic data generation process through the use of realistic affine registration

A Mixture of Experts (MoE) model was used for the resolution enhancement task. The model consisted of seven expert networks, each trained on a specific resolution range (from 1 to 7 mm$^3$). Considering the distribution of available data from the `volBrain` platform , we have decided to simplify the network selection based on volumetric resolution. Specifically, only one network (of the 7 trained ones) is selected based on the product of resolutions in each plane. The expert models were based on a modified version of the volumetric U-net (Ronneberger et al., 2015), incorporating a residual connection at the end of the network. The training process involved a composed loss function based on the combination of intensity-related metrics (mean absolute error (MAE), Structural Similarity Index Measure (SSIM) (Wang et al., 2004), gradient MAE loss) and a structure-related metric (correlation coefficient). A transfer learning approach was used to train the seven networks starting from the 1 mm$^3$ network.

## 3. Experiments and results

We compared our method with SynthSR using 45 cases of the third dataset (5 cases per resolution). For each case, the low-resolution counterpart was generated by randomly selecting the inverse affine transform for each of the seven resolutions from the `volBrain` dataset. The results, shown in Table 1, indicate that SynthSR demonstrates relatively constant performance intensity-related metrics (PSNR) across resolutions due to the clipping process performed to the output, where intensities are mapped to intensities of standard

MP-RAGE contrast. However, our expert models leverage intrinsic patterns of different resolutions. While SynthSR performs both super-resolution and image synthesis (maps the anatomy to a specific intensity range independent from the input intensity range), our method focuses solely on super-resolution, resulting in more coherent intensity-based metrics. Using the intensity-independent correlation coefficient, our method showed significant gains in plausibility and detail compared to SynthSR.

Table 1: Comparison of PSNR, Correlation and DICE metrics between SynthSR and our experts' models.

| Resolution | 1 mm$^3$ | 2 mm$^3$ | 3 mm$^3$ | 4 mm$^3$ | 5 mm$^3$ | 6 mm$^3$ | 7 mm$^3$ | Average |
|---|---|---|---|---|---|---|---|---|
| **Model name** | **PSNR** | | | | | | | |
| *SynthSR* | 18.2456 | 18.3044 | 18.0438 | 18.0955 | 17.996 | 17.9786 | 17.6018 | 18.038 |
| *Experts (7 models)* | **38.4631** | **36.0546** | **29.1123** | **27.8955** | **27.1152** | **26.5342** | **26.7493** | **30.2749** |
| **Model name** | **Correlation** | | | | | | | |
| *SynthSR* | 0.7268 | 0.7245 | 0.7009 | 0.6893 | 0.6797 | 0.6723 | 0.633 | 0.6895 |
| *Experts (7 models)* | **0.9937** | **0.988** | **0.963** | **0.9503** | **0.9455** | **0.9365** | **0.9352** | **0.9589** |
| **Model name** | **DICE** | | | | | | | |
| *None (from affine)* | 0.9904 | 0.9726 | 0.9237 | 0.8719 | 0.7678 | 0.7282 | 0.7128 | 0.8524 |
| *1 Network* | 0.9880 | 0.9718 | 0.923 | 0.8713 | 0.7631 | 0.7242 | 0.7066 | 0.8497 |
| *SynthSR* | 0.9049 | 0.9002 | 0.8791 | 0.8533 | 0.8096 | 0.7965 | 0.7648 | 0.8441 |
| *Experts (7 models)* | **0.9907** | **0.9734** | **0.9267** | **0.9021** | **0.8692** | **0.8568** | **0.8208** | **0.9057** |

Resolution enhancement is commonly used to improve the performance of segmentation networks. To evaluate our models, we measure the impact of using the SynthSeg segmentation method (Billot et al., 2023), which can segment brain scans with different contrasts and resolutions. The segmentation accuracy is evaluated using 30 labels from SynthSeg on high-resolution HCP data. The results, presented in Table 1, compare the segmentation outcomes of low-resolution images in MNI space without any model, our model trained on all resolutions simultaneously, SynthSR and our MOE. The expert models notably enhance the DICE compared to segmentations derived from original low-resolution data and a single generalistic model such as our own model or SynthSR. This underscores the impact of input data resolution on subsequent segmentation networks and highlights how seven expert networks specializing in various intrinsic patterns of volumetric resolutions can contribute to improved results.

## 4. Discussion and Conclusion

Our study examines expert models' effectiveness in improving brain imaging data resolution across various resolutions. Comparisons with general methods like SynthSR show significant improvements in PSNR and correlation coefficients. We have demonstrated that realistic data generation techniques enhance the model's performance with low-resolution data. The evaluation of segmentation performance, measured by the DICE coefficient, further underscores the critical role of resolution in influencing downstream tasks. Our approach consistently improves key performance metrics in super-resolving brain imaging data in MNI space, demonstrating utility across different current methods and modalities.

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
