# OpenReview forum: "REMIX: Resolution Enhancement through Mixture of Experts"
_MIDL.io/2024/Short_Papers — MIDL 2024 Short Papers_

### Official Review · Reviewer_irj5 · 2024-04-16

**Confidence:** 5
**Final Rating:** 5

**Review:**

Summary:
This paper presents an interesting method to perform realistic resolution augmentation, since CNNs are known to be fragile to changes in input resolution. This simple but sound method relies on affine transforms obtained by registering low resolution clinical scans to the MNI template. The resulting affine transforms are then inverted and applied to high resolution data to obtain paired high-low resolution images to train a super-resolution network.

Weaknesses:
- I think a baseline is missing, which would be to apply the resolution augmentation strategy of SynthSR to the real training dataset.
- Can the authors comment on the use of the correlation loss since correlation is already included in the SSIM metric ?

---

### Decision · Program_Chairs · 2024-04-26

Accept